# Búsqueda de vecindad variable para el problema de localización de instalaciones sin capacidad

**Lucas Martín-García**
Dpto. de Informática y Estadística
Universidad Rey Juan Carlos
Móstoles, España
lucas.martin@urjc.es

**Isaac Lozano-Osorio**
Dpto. de Informática y Estadística
Universidad Rey Juan Carlos
Móstoles, España
isaac.lozano@urjc.es

**J. Manuel Colmenar**
Dpto. de Informática y Estadística
Universidad Rey Juan Carlos
Móstoles, España
josemanuel.colmenar@urjc.es

**Belén Melián-Batista**
Dpto. de Ingeniería Informática y de Sistemas
Universidad de La Laguna
La Laguna, España
mbmelian@ull.edu.com

## Abstract

La localización de instalaciones sin capacidad constituye un problema relevante en ámbitos como la logística, la distribución de recursos y la planificación de redes de telecomunicaciones. Dados un conjunto de posibles instalaciones y un conjunto de clientes, el problema consiste en seleccionar un indeterminado número de instalaciones a abrir para servir a todos los clientes con el objetivo de minimizar los costes de apertura y asignación. Debido a su clasificación como problema $\mathcal{NP}$-difícil, la obtención de soluciones exactas se vuelve inabordable a gran escala, obligando a explorar métodos aproximados y metaheurísticas. En el marco de este problema, inicialmente se desarrollaron diversas aproximaciones exactas basadas. Sin embargo, más recientemente han cobrado relevancia los algoritmos aproximados y metaheurísticos, que logran soluciones de alta calidad con un coste computacional reducido. En este trabajo se propone una estrategia de búsqueda de vecindad variable que, mediante la apertura y cierre de instalaciones, equilibra la intensificación y la diversificación de la búsqueda. Los resultados experimentales sobre distintos conjuntos de instancias muestran que la propuesta logra soluciones de alta calidad muy próximas a las óptimas pero con un tiempo de ejecución reducido, comparándose con otros algoritmos del estado del arte.

## 1. Introducción

La familia de problemas de localización de instalaciones (*Facility Location Problems*, FLP) engloba un variado conjunto de problemas de optimización en los que se busca seleccionar la mejor configuración de instalaciones para optimizar objetivos de coste, tiempo o calidad de servicio en diferentes contextos, incluyendo logística, telecomunicaciones, y distribución de recursos [7]. Dentro de esta familia, el problema de localización de instalaciones sin capacidad (*Uncapacitated Facility Location Problem*, UFLP) se ha consolidado como uno de los más estudiados debido a su importancia en la teoría de la localización y a su amplia aplicabilidad en escenarios reales, como por ejemplo la distribución de redes ferroviarias y de carreteras [14].

A lo largo de la segunda mitad del siglo XX, la resolución de este problema se abordó a través de métodos exactos [8, 2, 10]. Estas propuestas constituyeron la primera generación de algoritmos capaces de resolver instancias de tamaño moderado y ofrecieron una perspectiva sólida para la

formulación del problema. Más adelante, se atacó el problema utilizando métodos aproximados que permitieron abordar instancias de mayor tamaño, destacando la búsqueda local voraz con adaptación progresiva de costes [5]. Finalmente, el problema se ha abordado utilizando diferentes metaheurísticas como algoritmos genéticos [17], recocido simulado [3] y diferentes algoritmos evolutivos [16, 1].

Actualmente, uno de los trabajos más relevantes, que se tomará como referencia para comparar la propuesta de este artículo, es el algoritmo *Enhanced Group Theory-Based Optimization Algorithm* (EGTOA) [21], una variante evolucionada del *Group Theory-Based Optimization Algorithm* (GTOA) [13] que introduce mejoras clave para optimizar el rendimiento en el UFLP. Los resultados experimentales sobre 15 instancias del OR-Library muestran que EGTOA supera a 16 algoritmos de la literatura en términos de calidad de solución y convergencia.

En este trabajo, se presenta una estrategia basada en la Búsqueda de Vecindad Variable (*Variable Neighborhood Search*, VNS) con movimientos de apertura y cierre de instalaciones. Los resultados experimentales muestran que este enfoque logra soluciones competitivas en el conjunto de instancias estudiado. En concreto, las principales contribuciones de este trabajo son las siguientes: (i) adaptación de una versión básica de VNS al problema UFLP, (ii) análisis de su rendimiento en un conjunto de instancias ampliado, y (iii) comparación con diversos métodos exactos y aproximados del estado del arte, evidenciando las fortalezas de la propuesta en cuanto a calidad y tiempos de ejecución.

El resto del artículo se estructura de la siguiente manera. En la Sección 2 se describe en detalle la formulación matemática del UFLP y la representación de soluciones propuesta en este trabajo. En la Sección 3 se describe la adaptación de la metodología VNS propuesta, explicando y analizando los experimentos en la Sección 4. Finalmente, se incluyen las conclusiones del trabajo en la Sección 5.

## 2. Descripción del problema

El problema de localización de instalaciones sin capacidad (UFLP) consiste en, dado un conjunto de posibles instalaciones, seleccionar aquel subconjunto de ellas que se abrirían para dar servicio a un conjunto de clientes, minimizando el coste total. Este coste incluye tanto los costes de apertura de instalaciones como los costes de servicio de los clientes. Cabe destacar que el número de instalaciones a abrir no está inicialmente determinado.

Más formalmente, dado un conjunto de $M$ clientes $A = \{a_1, a_2, \ldots, a_M\}$ y un conjunto de $N$ instalaciones potenciales $B = \{b_1, b_2, \ldots, b_N\}$, se define la matriz de costes $C = [c_{ij}]_{M \times N}$ donde $c_{ij}$ representa el coste de servir al cliente $a_i$ desde la instalación $b_j$. Además, cada instalación $b_j$ tiene un coste fijo de apertura $d_j$.

La formulación del problema requiere la introducción de variables de decisión binarias para modelar la apertura de instalaciones y la asignación de clientes a estas. En este sentido, para cada instalación $b_j$ (con $j = 1, \ldots, N$) se define la variable $x_j$, que toma el valor 1 si la instalación $b_j$ se abre, y 0 en caso contrario. De manera análoga, para cada cliente $a_i$ (con $i = 1, \ldots, M$) y para cada instalación $b_j$ (con $j = 1, \ldots, N$) se introduce la variable $y_{ij}$. Esta variable toma el valor 1 si el cliente $a_i$ es asignado a la instalación $b_j$, y 0 en caso contrario. La Ecuación (1) muestra la formulación para la función objetivo, mientras que las ecuaciones (2) a (4) aseguran que cada cliente es asignado a exactamente una instalación abierta.

$$\text{mín} \quad \sum_{j=1}^{N} d_j x_j + \sum_{i=1}^{M} \sum_{j=1}^{N} c_{ij} y_{ij} \tag{1}$$

$$\text{sujeto a} \quad \sum_{j=1}^{N} y_{ij} = 1, \quad \forall i \in \{1, 2, ..., N\} \tag{2}$$

$$y_{ij} - x_j \leq 0, \quad \forall i \in \{1, 2, ..., N\}, \forall j \in \{1, 2, ..., M\} \tag{3}$$

$$y_{ij}, x_j \in \{0, 1\}, \quad \forall i \in \{1, 2, ..., N\}, \forall j \in \{1, 2, ..., M\}. \tag{4}$$

Dada una configuración de instalaciones abiertas y cerradas para una instancia, la asignación de clientes óptima, que obtiene el mínimo valor de la función objetivo para esa configuración, se obtiene asignando cada cliente a la instalación abierta cuyo coste de asignación para ese cliente es el menor.

De esta forma, una vez establecidas las instalaciones que se abren y las que se cierran, la asignación siempre se puede realizar de manera eficiente, escogiendo para cada cliente la instalación abierta con el menor coste de asignación. Por lo tanto, toda solución del problema se puede representar mediante las instalaciones abiertas y las cerradas sin necesidad de representar la asignación de clientes. En este trabajo, se representa una solución $S$ como el conjunto de instalaciones abiertas, dejando fuera del conjunto las instalaciones cerradas, es decir, $S = \{a_i : a_i \in A \text{ está abierta }\}$. De la misma manera, el valor de coste para una solución $S$ determinado por la función objetivo definida en la Ecuación (1) se denotará como $\mathcal{F}(S)$.

A modo de ilustración, en la Figura 1 se muestra una instancia con cuatro clientes representados por círculos azules y tres instalaciones potenciales $b_1$, $b_2$ y $b_3$ representadas por cuadrados. Cada instalación conlleva un coste de apertura de 4 unidades, mientras que los costes de asignación para cada cliente (representados en las aristas) corresponden a las distancias euclidianas entre cliente e instalación.

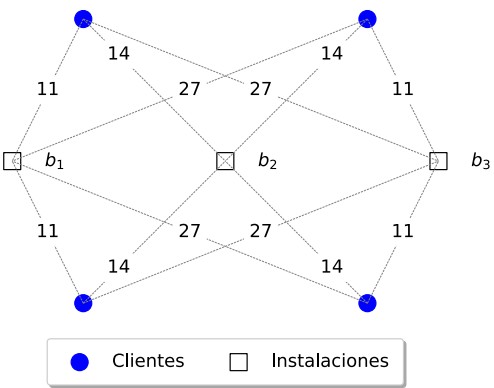

Figura 1: Instancia del UFLP con cuatro clientes (círculos azules) y tres instalaciones potenciales (cuadrados vacíos). Los valores sobre las aristas representan el coste de servir a cada cliente desde la correspondiente instalación.

Para esta instancia existen varias soluciones posibles. La primera configuración que se podría considerar, con el objetivo de reducir los costes de apertura, consiste en abrir únicamente $b_2$, dado que, en promedio, se encuentra a menor distancia de todos los clientes, tal como se aprecia en la Figura 2a. Esta solución, $S_1 = \{b_2\}$, conlleva un coste de asignación con valor 14 por cada uno de los cuatro clientes (56 en total), al que se suma el coste de apertura (4), resultando en un valor de función objetivo $\mathcal{F}(S_1) = 60$. La Figura 2b muestra la solución $S_2 = \{b_1, b_3\}$, donde aunque el coste de apertura se duplica, la suma de los costes de asignación se reduce a 44, dando lugar a $\mathcal{F}(S_2) = 52$, que es la solución óptima para este ejemplo. Nótese que si se opta por abrir las tres instalaciones, se incrementa el gasto fijo hasta 12, sin beneficio adicional, ya que los clientes se asignarían igualmente a las dos instalaciones laterales, que son las más cercanas, dejando a la instalación central sin utilizar. Este ejemplo refleja la complejidad inherente de este problema, pues, al no estar acotado el número de instalaciones que pueden abrirse, aumenta significativamente la cantidad de configuraciones factibles, incluso en una instancia con solo tres posibles instalaciones.

## 3. Propuesta algorítmica

El problema UFLP es un problema de optimización combinatoria clasificado como $\mathcal{NP}$-difícil [21]. Debido a su complejidad, los enfoques exactos resultan ineficientes para instancias de gran tamaño. Por ello, en este trabajo se propone un algoritmo basado en la metaheurística Búsqueda de Vecindad Variable (*Variable Neighborhood Search*, VNS), con el objetivo de obtener soluciones de alta calidad en tiempos de cómputo razonables.

VNS es una metaheurística basada en la exploración sistemática de múltiples vecindarios y apoyada en un mecanismo de perturbación que permite escapar de óptimos locales [20, 12].

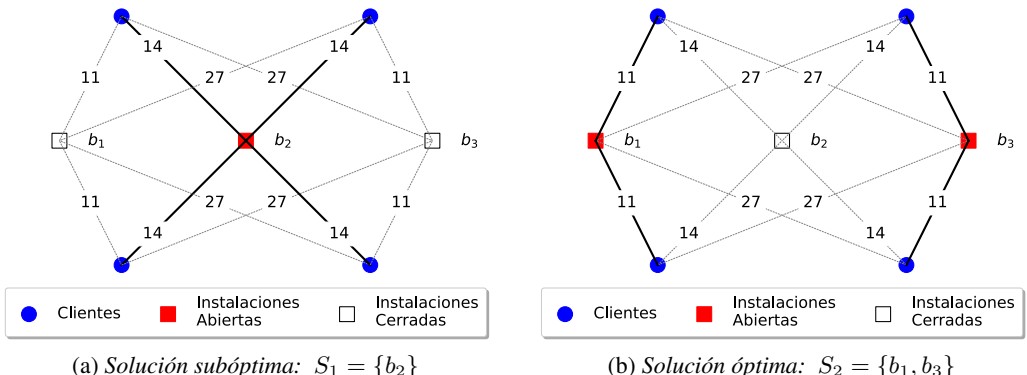

(a) *Solución subóptima:* $S_1 = \{b_2\}$        (b) *Solución óptima:* $S_2 = \{b_1, b_3\}$

Figura 2: Soluciones alternativas para la instancia de ejemplo.

Esta metaheurística ha sido ampliamente utilizada en problemas de optimización combinatoria, demostrando su efectividad en la obtención de soluciones de alta calidad. Por ejemplo, ha sido aplicada a diferentes problemas de la familia de la *p*-mediana como el *p-Median Problem* (PMP), donde se ha implementado una estrategia de VNS paralelizada, mostrando que esta metaheurística es altamente eficiente en problemas de asignación y localización de clientes a instalaciones [11]. Otro ejemplo de la misma familia es el *Capacitated p-Median Problem* (CPMP), obteniendo soluciones competitivas mediante una versión optimizada de VNS [9]. Más recientemente, se ha propuesto una variante de VNS al problema de la mediana con instalaciones interconectadas [18]. Finalmente, VNS también se ha aplicado a problemas de localización de instalaciones como el *Bus Terminal Location Problem* (BTLP), donde una variante paralelizada de VNS ha demostrado mejoras significativas en términos de calidad de solución y tiempo de cómputo [6].

En este trabajo se propone la aplicación de VNS al UFLP, aprovechando su capacidad para explorar el espacio de soluciones de manera eficiente y evitar la convergencia prematura a óptimos locales. En particular, este trabajo se enfoca en la variante *Basic* VNS (BVNS) [12]. BVNS combina cambios aleatorios (diversificación) y deterministas (intensificación) en la exploración de vecindades, lo que contribuye a mejorar la calidad de las soluciones y evitar el estancamiento en óptimos locales. En el Algoritmo 1 se presenta el pseudocódigo del algoritmo BVNS propuesto para abordar el UFLP.

---

**Algoritmo 1** BVNS($k_{\text{máx}}$,N)

---

1:   $S \leftarrow$ Constructivo(N)
2:   $S^* \leftarrow S$
3:   $k \leftarrow 0$
4:   **while** $k < k_{\text{máx}}$ **do**
5:      $S \leftarrow$ Shake(S,k)
6:      $S \leftarrow$ Mejora(S)
7:      **if** $\mathcal{F}(S) < \mathcal{F}(S^*)$ **then**
8:         $S^* \leftarrow S$
9:         $k \leftarrow 1$
10:     **else**
11:        $k \leftarrow k + 1$
12:     **end if**
13: **end while**
14: **return** $S^*$

---

El algoritmo BVNS($k_{\text{máx}}$,N) requiere como parámetro de entrada el número de instalaciones que se perturban como máximo ($k_{\text{máx}}$) y el número de instalaciones total (N). La ejecución comienza obteniendo una solución inicial a través de un método constructivo (línea 1, véase la Sección 3.1), que se convierte en la mejor solución actual, denotada como $S^*$. A continuación, el bucle principal (líneas 4–13) itera hasta que no se consiga mejorar la solución durante $k_{\text{máx}}$ iteraciones. En cada iteración, se perturba la solución actual mediante la función Shake (línea 5), que modifica $k$ instalaciones

de manera aleatoria para diversificar la búsqueda (véase la Sección 3.3). A continuación, se aplica la función `Mejora` (véase la Sección 3.2) sobre la solución $S$ (línea 6). Finalmente, se comprueba si la perturbación y la mejora han generado una solución mejor que la actual, reiniciando el valor de $k$ en caso afirmativo (líneas 7– 9) o incrementándolo en caso contrario. Además, cuanto mayor es la $k$, mayor será la perturbación generada en la solución. Después de explorar la máxima vecindad permitida, el algoritmo devuelve la mejor solución encontrada durante todo el proceso (línea 14).

## 3.1. Método constructivo

La fase constructiva del algoritmo consiste en generar una solución inicial $S$ de forma aleatoria, garantizando que al menos una instalación esté abierta para que la solución generada sea factible. Primero, se inicializa la solución como el conjunto vacío, es decir, con todas las instalaciones cerradas. Posteriormente, para cada instalación posible, se decide con probabilidad $0{,}50$ si se añade la instalación a la solución, es decir, si se abre la instalación.

En caso de que, tras procesar todas las instalaciones, $S$ permanezca vacío, se fuerza la apertura de una instalación seleccionada aleatoriamente. Con ello se evita la solución trivial de no abrir ninguna instalación y se garantiza la factibilidad del problema. Esta estrategia simple y rápida aporta diversidad en la solución inicial, lo que puede favorecer una mayor exploración del espacio de soluciones en etapas posteriores del algoritmo. Debido a la limitación de espacio y simplicidad del método, se omite el pseudocódigo correspondiente.

## 3.2. Método de mejora

En este trabajo, durante la fase de mejora del algoritmo propuesto se utiliza como movimiento la operación de cerrar instalaciones que se encuentran abiertas en la solución. Este movimiento que cierra instalaciones abiertas genera los vecindarios de las soluciones que se exploran en el algoritmo BVNS. En trabajos previos se ha comprobado que en la práctica este tipo de movimiento suele mejorar la eficiencia y la calidad de la búsqueda, tal y como se expone en [21]. En dicho trabajo, los autores justifican que, al cerrar instalaciones abiertas, se reduce el número total de emplazamientos activos y con ello los costes de apertura, promoviendo así soluciones de mayor calidad. Además, se subraya que mantener demasiadas instalaciones abiertas tiende a elevar el valor de la función objetivo, por lo que clausurar aquellas que no contribuyen positivamente mejora la calidad de la solución.

La función de mejora presentada en el Algoritmo 2 implementa una búsqueda local con la estrategia *best improvement* que utiliza el movimiento de cerrar instalaciones. Esta función recibe como entrada una solución factible $S$ y busca iterativamente reducir el valor de la función objetivo $\mathcal{F}$. Para ello, en la línea 1 se inicializa la variable *mejora*, que sirve como criterio de parada del bucle principal (línea 2). Este bucle se ejecuta mientras se haya conseguido mejorar la solución actual y termina cuando ya no se puede reducir la función objetivo de la solución mediante el movimiento de cerrar instalaciones. En cada iteración, se identifica la instalación $x$ cuyo cierre en $S$ produce la mayor reducción en la función objetivo (línea 4). A continuación, se construye una nueva solución $S'$ que no incluye a $x$. Si el valor de la función objetivo de la nueva solución $S'$ es mejor que la solución original $S$, se actualiza la solución y se señala que ha habido una mejora (líneas 6– 8). De lo contrario, no se realizan más cambios y la búsqueda local finaliza. En consecuencia, este procedimiento elimina progresivamente las instalaciones que no aportan una contribución positiva, refinando la solución hasta alcanzar un punto en el que no se puedan efectuar más mejoras.

Además de la estrategia de búsqueda local *best improvement* descrita anteriormente, también se ha implementado una variante basada en la estrategia *first improvement*. En esta versión, en lugar de evaluar todos los vecinos de la solución actual y seleccionar el mejor, los movimientos de cierre de instalaciones se exploran en un orden aleatorio y se selecciona el primero que mejora la solución actual. En cuanto se encuentra una instalación cuyo cierre mejora la función objetivo, la solución se actualiza inmediatamente y el proceso se reinicia con la nueva solución mejorada. Para adaptar el pseudocódigo de la búsqueda local *best improvement* a esta variante, basta con modificar la selección del vecino en cada iteración. En vez de determinar la instalación cuyo cierre genere la mayor mejora en la función objetivo, se considera una permutación aleatoria de las instalaciones abiertas y se exploran en el orden establecido hasta encontrar la primera mejora. Al encontrar una mejora, la iteración se detiene y la búsqueda se reinicia desde la nueva solución obtenida.

**Algoritmo 2** Mejora(S)

```
 1: mejora ← Verdadero
 2: while mejora do
 3:     mejora ← Falso
 4:     x ← arg mín_{i∈S} F(S \ {i})
 5:     S' ← S \ {x}
 6:     if F(S') < F(S) then
 7:         mejora ← Verdadero
 8:         S ← S'
 9:     end if
10: end while
11: return S
```

### 3.3. Método de perturbación

La función de perturbación descrita en el Algoritmo 3, tiene como objetivo realizar perturbaciones sobre una solución dada $S$ para desplazarse a un nuevo vecindario, una vez que la búsqueda local alcanza un óptimo local. Concretamente, en este trabajo se ha escogido la estrategia de abrir instalaciones como forma de perturbación, complementando así el movimiento de cerrar instalaciones utilizado en la búsqueda local. De este modo, al finalizar cada fase de mejora, permite alejarse de la vecindad explorada, añadiendo un subconjunto de instalaciones que estaban cerradas, en función del parámetro $k$ que indica cuántas instalaciones se abrirán.

**Algoritmo 3** Shake($S$,$k$)

```
 1: C ← { a_i : a_i ∈ A ∧ a_i ∉ S}
 2: C' ← SeleccionAleatoria(C,k)
 3: S ← S ∪ C'
 4: return S
```

En el pseudocódigo se muestra cómo la lista de candidatos $C$ (línea 1) almacena todas las instalaciones que están cerradas. A continuación, se obtienen $k$ instancias de manera aleatoria, considerando que el máximo a abrir en cada perturbación es $min(k, |C|)$ (línea 2) para diversificar las aperturas. Finalmente, se añaden las instalaciones seleccionadas a la solución actual (línea 3), que se devuelve al finalizar el algoritmo (línea 4).

## 4. Resultados experimentales

Se presenta a continuación el conjunto de experimentos realizados para evaluar el algoritmo propuesto y compararlo con el algoritmo evolutivo EGTOA y el modelo exacto, considerados como referencia en el estado del arte. Todos los experimentos se han realizado en un servidor AMD EPYC 7643 de 32 núcleos con 32 GB de memoria RAM. El algoritmo propuesto se ha implementado usando el lenguaje de programación Java 21.

En este trabajo se han utilizado las instancias de la literatura. Cabe destacar que se han ampliado estas instancias con dos conjuntos de datos de problemas directamente relacionados. Estos conjuntos de datos se han evaluado sobre el mismo entorno experimental con el objetivo de realizar una comparación robusta entre los algoritmos y analizar su desempeño con distintos parámetros del problema. Toda la información relativa a las características de los conjuntos de instancias utilizados se encuentra en la Tabla 1. El primer conjunto de instancias está formado por las instancias *Cap*, ampliamente utilizadas en esta familia de problemas de localización de instalaciones, y publicadas en *OR-Library* [4][1]. Dado que este conjunto de instancias es reducido, se ha optado por explorar conjuntos más extensos de instancias reconocidas en la literatura. De este modo, se han añadido a la comparación los conjuntos *K90* y *G700* [14]. Estas instancias han sido generadas a partir de la red de vías férreas y carreteras de la República Eslovaca y se encuentran disponibles en abierto[2]. El

---

[1] https://people.brunel.ac.uk/~mastjjb/jeb/orlib/uncapinfo.html
[2] http://frdsa.uniza.sk/~buzna/supplement

conjunto de instancias *K90* consta de 90 instancias individuales; sin embargo, tres de ellas presentan valores de coste negativos, por lo que han sido excluidas del análisis para mantener la consistencia con el resto de las instancias.

Tabla 1: Descripción de los conjuntos de instancias estudiados.

| Nombre | Número | Instalaciones | Clientes | Referencias |
|--------|--------|---------------|----------|-------------|
| *Cap* | 15 | 16 - 100 | 50 - 1000 | [4] |
| *K90* | 87 | 45 - 457 | 457 | [14] |
| *G700* | 700 | 100 - 1000 | 2906 | [14] |

El algoritmo EGTOA se ha ejecutado sobre los nuevos conjuntos de instancias gracias a que sus autores publicaron el código fuente junto con el artículo [21], lo que ha permitido su adaptación y aplicación en este estudio. Por otro lado, el modelo exacto se ha implementado en Gurobi, donde se ha utilizado el modelo matemático del problema.

Las tablas de resultados de los experimentos presentan cuatro métricas clave para evaluar el desempeño de los algoritmos en distintas instancias del problema. La métrica denotada como $\mathcal{F}$ refleja el valor promedio obtenido por cada algoritmo, donde un menor valor indica una mejor solución en términos del criterio de optimización. La métrica *Tiempo (s)* representa el tiempo promedio de cómputo requerido para ejecutar cada algoritmo, representado en segundos. La métrica *Gap ( %)* mide la diferencia porcentual promedio entre las soluciones encontradas por el algoritmo y las soluciones óptimas, proporcionando una referencia del error relativo. Finalmente, la métrica *# Óptimos* indica el número de instancias en las que cada algoritmo ha encontrado la solución óptima.

Los experimentos se estructuran en dos fases: una fase preliminar y una fase de comparación final. La fase preliminar consiste en la realización de pruebas para determinar los mejores valores para los parámetros que configuran el algoritmo propuesto. Seguidamente, la fase final evalúa el rendimiento de la mejor configuración seleccionada, comparándola con los otros métodos explicados anteriormente.

## 4.1. Experimentos preliminares

La fase de experimentación preliminar se llevó a cabo sobre un subconjunto representativo de las instancias. Para ello, se seleccionó aleatoriamente el 20 % de las instancias de cada conjunto, garantizando una distribución equitativa. Cabe destacar que los conjuntos de instancias presentan diferencias en cuanto al origen de los datos, la cantidad de instancias, el tamaño de los problemas y la distribución de los valores de coste. Por esta razón, la selección del subconjunto preliminar se realizó mediante un muestreo estratificado, asegurando que cada conjunto estuviera representado proporcionalmente en la muestra y preservando la diversidad en los datos.

El objetivo principal de esta fase preliminar es establecer los mejores valores para los parámetros del algoritmo propuesto. Por ese motivo, se ha ejecutado el algoritmo BVNS con las dos búsquedas locales *best improvement* y *first improvement* para cada uno de los siguientes valores: $K = \{0{,}1; 0{,}2; 0{,}3; 0{,}4; 0{,}5\}$. Nótese que en el Algoritmo 1 se recibe como parámetro $k_{max} = \lceil N \cdot K \rceil$. Los resultados obtenidos en este experimento se muestran en la Tabla 2.

Analizando la tabla, se puede observar que la estrategia *best improvement* obtiene mejores resultados en todas las métricas utilizadas. Cabe destacar que a mayor valor de $K$, generalmente se obtienen mejores valores de $\mathcal{F}$, siendo $K = 0{,}5$ en *first improvement* y $K = 0{,}4$ en *best improvement* los valores que obtienen mejores resultados de función de coste y *gap* para cada una de las estrategias. En cuanto al tiempo de ejecución, el valor de $K$ es directamente proporcional al tiempo empleado. Esto se debe a que el valor de $K$ afecta directamente al tamaño de la perturbación provocada en la solución, aumentando el tiempo de cómputo requerido por el algoritmo. En la Tabla 2 se pueden observar unos porcentajes de desviación respecto de los valores óptimos cercanos a cero, indicando un buen desempeño del algoritmo en este subconjunto de instancias. Finalmente, la búsqueda local *best improvement* obtiene más del doble de soluciones óptimas frente a la búsqueda local *first improvement* para todos los valores de $K$, mostrando un mejor rendimiento del algoritmo con este tipo de búsqueda. El mejor resultado para esta métrica se consigue con el valor $K = 0{,}5$ obteniendo 89 soluciones

Tabla 2: Resultados del algoritmo BVNS según la estrategia de búsqueda local y valores de $K$.

| Búsqueda Local | $K$ | $\mathcal{F}$ | Tiempo (s) | Gap ( %) | # Óptimos |
|---|---|---|---|---|---|
| | 0,1 | 3 201 593,58 | 11,66 | 0,8096 | 25 |
| | 0,2 | 3 191 440,99 | 13,90 | 0,6698 | 29 |
| *First improvement* | 0,3 | 3 185 544,26 | 17,43 | 0,6613 | 30 |
| | 0,4 | 3 187 689,18 | 22,45 | 0,6555 | 31 |
| | 0,5 | 3 177 980,34 | 30,48 | 0,5416 | 39 |
| | 0,1 | 3 176 264,31 | **2,95** | 0,1385 | 55 |
| | 0,2 | 3 172 301,69 | 5,89 | 0,0470 | 80 |
| *Best improvement* | 0,3 | 3 174 707,54 | 10,02 | 0,0619 | 77 |
| | 0,4 | **3 170 593,02** | 15,94 | **0,0355** | 87 |
| | 0,5 | 3 171 236,10 | 23,22 | 0,0385 | **89** |

óptimas de las 161 instancias del subconjunto preliminar, aunque la diferencia con el segundo mejor resultado obtenido con el valor $K = 0{,}4$ es de solo 2 óptimos.

Tras analizar los resultados obtenidos, se puede concluir que la mejor configuración para el algoritmo BVNS es la que emplea la búsqueda local *best improvement* y el valor $K = 0{,}4$. A pesar de que con esta configuración el algoritmo obtiene dos óptimos menos que con el valor $K = 0{,}5$, en el resto de métricas se obtienen mejores resultados, con un tiempo de ejecución menor y un valor de función objetivo y porcentaje de desviación inferiores. Por tanto, esta será la configuración elegida para la comparativa con el estado del arte.

## 4.2. Comparación con el estado del arte

Una vez establecida la configuración del algoritmo propuesto, se ha comparado su rendimiento con el algoritmo del estado del arte EGTOA y el algoritmo exacto implementado a partir del modelo matemático. La Tabla 3 muestra los resultados promedio de los tres algoritmos comparados en cada uno de los conjuntos de instancias estudiados en este trabajo. Como se puede apreciar, se han incluido los resultados agregados al final de la tabla.

Tabla 3: Resultados de los algoritmos EGTOA, BVNS y el modelo exacto para las instancias estudiadas.

| Instancias | Algoritmo | $\mathcal{F}$ | Tiempo (s) | Gap ( %) | # Óptimos |
|---|---|---|---|---|---|
| | EGTOA | 3 502 545,59 | 258,931 | 0,0000 | 15 |
| *Cap* | BVNS | 3 509 998,78 | 0,027 | 0,0823 | 12 |
| | Modelo Exacto | 3 502 545,59 | 0,802 | 0,0000 | 15 |
| | EGTOA | 3 069 045,61 | 1 405,238 | 12,1286 | 0 |
| *K90* | BVNS | 2 768 151,54 | 0,321 | 0,0891 | 28 |
| | Modelo Exacto | 2 765 298,57 | 2,642 | 0,0000 | 87 |
| | EGTOA | 27 202 096,73 | 3 597,980 | 174,8163 | 23 |
| *G700* | BVNS | 2 947 040,14 | 16,881 | 0,0246 | 435 |
| | Modelo Exacto | 2 943 809,00 | 62,861 | 0,0000 | 700 |
| | EGTOA | 24 140 913,79 | 1 236,663 | 153,8985 | 38 |
| Agregado | BVNS | 2 938 163,67 | **14,769** | 0,0327 | 475 |
| | Modelo Exacto | **2 934 894,58** | 55,168 | **0,0000** | **802** |

Los resultados agregados muestran que la propuesta BVNS ofrece un buen rendimiento en comparación con EGTOA. Observando la métrica $\mathcal{F}$, el algoritmo EGTOA se aleja en gran medida de los otros dos algoritmos con un resultado más de 10 veces mayor (24 140 913,79) que el modelo exacto (2 934 894,58), mientras que el BVNS obtiene un resultado cercano (2 938 163,67). Aunque BVNS no alcanza la solución óptima para todas las instancias, presenta una desviación muy cercana a cero (0,0327 %) y un tiempo de ejecución (14,769 segundos) que es más de tres veces menor que el del

modelo exacto (55,168 segundos). Esto indica un destacado balance entre calidad, escalabilidad y eficiencia computacional del algoritmo BVNS propuesto.

En el subconjunto *Cap*, tanto EGTOA como el modelo exacto obtienen todas las soluciones óptimas, aunque EGTOA requiere 258,931 segundos y el modelo exacto 0,802 segundos. En contraste, BVNS obtiene una desviación de 0,0823 % en tan solo 0,033 segundos, demostrando una notable reducción en el tiempo de ejecución con margen para lograr conseguir los tres óptimos restantes.

En el conjunto *K90*, el modelo exacto emplea 2,642 segundos de ejecución en promedio, mientras que BVNS registra una desviación de 0,0891 % en solo 0,464 segundos y obtiene 28 óptimos. Por otro lado, EGTOA muestra un desempeño inferior, con una desviación de 12,1286 % y un tiempo elevado de 1 405,238 segundos sin llegar a alcanzar ningún valor óptimo.

Finalmente, en el subconjunto *G700*, el modelo exacto obtiene todas las soluciones óptimas en 62,861 segundos, pero BVNS destaca al lograr una desviación de 0,0246 % en 16,881 segundos, alcanzando 435 óptimos. En este caso, EGTOA presenta una desviación muy elevada (174,8163 %) y un tiempo muy superior (3 597,980 segundos), demostrando su peor desempeño en instancias de mayor complejidad.

Cabe destacar que, aunque el algoritmo EGTOA ha sido evaluado solo sobre las instancias básicas Cap en el artículo relacionado, obteniendo muy buenos resultados, el tiempo de ejecución de este algoritmo (258,931 segundos) es más de 300 veces superior al tiempo del modelo exacto (0,0823 segundos), ambos ejecutados sobre las mismas instancias y en la misma máquina. Además, al aplicar este algoritmo a conjuntos de mayor tamaño como *K90* y *G700* se obtiene un desempeño peor que en el conjunto Cap. En contraposición, el algoritmo BVNS demuestra capacidad para resolver instancias más complejas y de mayor tamaño con una eficiencia y calidad de resolución que, con un proceso de optimización más completo y con métodos más elaborados, lo posicionarían como una alternativa competitiva frente al modelo exacto y al algoritmo EGTOA.

## 5.  Conclusiones

En este trabajo se ha propuesto un algoritmo basado en *Basic Variable Neighborhood Search* (BVNS) para resolver el problema de localización de instalaciones sin capacidad (UFLP). La metodología combina una búsqueda local basada en un movimiento de cierre de instalaciones y un procedimiento de perturbación que abre nuevas instalaciones, logrando un equilibrio entre intensificación y diversificación.

Los experimentos realizados sobre tres conjuntos de instancias (*Cap*, *K90* y *G700*) muestran que la propuesta BVNS obtiene un número mayor de soluciones óptimas frente al algoritmo evolutivo previo (475 de BVNS frente a 38 de EGTOA). Adicionalmente, la propuesta obtiene una desviación de 0,0327 % en comparación con las soluciones óptimas verificadas por el modelo exacto, al tiempo que reduce los tiempos de cómputo en comparación con los métodos del estado del arte. En particular, BVNS destaca en las instancias más grandes, ofreciendo desviaciones muy pequeñas respecto del óptimo con un esfuerzo computacional reducido, a diferencia del algoritmo EGTOA, que obtiene una desviación cuatro órdenes de magnitud mayor que BVNS con un tiempo de ejecución muy superior en los conjuntos de instancias *K90* y *G700*.

Como trabajo futuro se plantea incorporar técnicas de aprendizaje automático. En particular, enfoques basados en aprendizaje por refuerzo, para la selección dinámica de vecindarios y la guía de la búsqueda, siguiendo la línea de trabajos recientes como [15, 19]. Con ello, se pretende potenciar la capacidad de exploración del algoritmo y mejorar aún más su desempeño en escenarios complejos y de mayor escala.

## Agradecimientos y declaración de financiación

Este trabajo ha sido parcialmente financiado por el "Ministerio de Ciencia, Innovación y Universidades (MCIN/AEI/10.13039/501100011033/FEDER, UE) bajo las subvenciones ref. RED2022-134480-T y PID2021-125709OA-C22, y por el "Ministerio para la Transformación Digital y de la Función Pública" mediante Concesión TSI-100930-2023-3 y de la Comunidad Autónoma de Madrid, con el proyecto CIRMA-CM (referencia TEC-2024/COM-404).

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
