# OpenReview forum: "Búsqueda de vecindad variable para el problema de localización de instalaciones sin capacidad"
_MAEB/2025/Congreso — MAEB 2025_

### Official Review · Reviewer_ssBh · 2025-03-12
**Búsqueda de vecindad variable para el problema de localización de instalaciones sin capacidad**

**Rating:** 5
**Confidence:** 2

**Review:**

Este trabajo propone un algoritmo basado en Basic Variable Neighborhood Search (BVNS) para resolver el problema de localización de instalaciones sin capacidad (UFLP). La metodología propuesta pretende equilibrar intensificación y diversificación mediante una búsqueda local que permite cerrar instalaciones y un procedimiento de perturbación que abre nuevas. Los experimentos en tres conjuntos de instancias (Cap, K90 y G700) muestran que BVNS obtiene 475 soluciones óptimas frente a las 38 del algoritmo evolutivo EGTOA, logra una desviación mínima respecto al óptimo y reduce los tiempos de cálculo. Esto se aprecia especialmente en instancias grandes.

El artículo está bien estructurado y redactado con claridad. Presenta un algoritmo que arroja buenos resultados frente a las soluciones actuales, particularmente a medida que el tamaño del problema aumenta.

Quisiera apuntar una cuestión que me parece extraña, y es que en la formulación del problema a resolver, parece que la decisión (que proporciona el algoritmo) implique solamente la ubicación de las instalaciones y que sean dependientes de la de los clientes, que parecen ser siempre fijas.
Me cuesta pensar en situaciones diferentes a empresas de reparto/distribución donde "clientes" se puede referir a población de tamaño mediano/grande. Sugeriría a los autores que incluyeran un sencillo ejemplo con el que ilustrar en qué tipo de situaciones aparece este problema.

---

### Official Review · Reviewer_o25Q · 2025-03-17
**El artículo presenta un algoritmo basado en Basic Variable Neighborhood Search (BVNS)  para resolver el problema de localización de instalaciones sin capacidad (UFLP).**

**Rating:** 5
**Confidence:** 4

**Review:**

El artículo presenta un algoritmo basado en Basic Variable Neighborhood Search (BVNS) para resolver el problema de localización de instalaciones sin capacidad (UFLP). La propuesta equilibra intensificación y diversificación mediante una búsqueda local. Los experimentos en tres conjuntos de instancias muestran que BVNS supera ampliamente a EGTOA logrando muchas mas soluciones óptimas, con una desviación mínima respecto al óptimo y tiempos de ejecución más bajos. La metodología está bien estructurada y los resultados demuestran que BVNS es una opción competitiva frente a otras técnicas del estado del arte. Aunque BVNS se compara con EGTOA, no se analiza frente a métodos exactos más sofisticados o enfoques heurísticos alternativos, lo que podría proporcionar una visión más completa de su rendimiento.

Finalmente, la estrategia de mejora del algoritmo se basa únicamente en cerrar instalaciones para reducir costes, sin considerar una posible reasignación óptima de clientes antes del cierre, lo que podría mejorar la calidad de las soluciones obtenidas. Con estas adiciones, el artículo mejoraría para el futuro y aportaría una contribución más sólida de optimización en este tipo de problemas.

---

### Official Review · Reviewer_LiDB · 2025-03-18
**Búsqueda de vecindad variable para el problema de localización de instalaciones sin capacidad**

**Rating:** 2
**Confidence:** 4

**Review:**

This paper is about the Uncapacitated Facility Location Problem and proposes a basic variable neighbourhood search algorithm. The algorithm is very straightforward and I'm mainly focusing on the experimental results section.

There I have two mayor difficulties.

1) The first is the preliminary experiments. The difficulty is that the computation times are not the same and it's not clear that the one that is chosen is the best one. Honestly, this only can be done by imposing each experiment the same computational time.

2) The final results. It is not clear that one would chose the BVNS over the Gurobi model, the exact one. There are two issues. The first one is that the results of the exact algorithm is proven optimal and this only in 4 times the differences with respect to the results of the BVNS. Why does one not take this additional time? Anyway, if you run it, one does not know if the results are optimal or not. The second is that the Gurobi takes some time to proof that it is the optimal after having found it. How high is this time? One would have to test it, but essentially it it gives the Gurobi even more advantages over the time that is given there.

---

### Decision · Program_Chairs · 2025-03-20

Accept